# NEK Family Review and Correlations with Patient Survival Outcomes in Various Cancer Types

**DOI:** 10.3390/cancers15072067

**Published:** 2023-03-30

**Authors:** Khoa Nguyen, Julia Boehling, Minh N. Tran, Thomas Cheng, Andrew Rivera, Bridgette M. Collins-Burow, Sean B. Lee, David H. Drewry, Matthew E. Burow

**Affiliations:** 1Department of Medicine, Tulane University School of Medicine, New Orleans, LA 70112, USA; 2Department of Pathology and Laboratory Medicine, Tulane University School of Medicine, New Orleans, LA 70112, USA; 3UNC Lineberger Comprehensive Cancer Center, UNC Eshelman School of Pharmacy, University of North Carolina at Chapel Hill, Chapel Hill, NC 27599, USA

**Keywords:** NEK family, kinase, cancer signaling, KMPlot, cosmic mutation, bioinformatics, illuminating the druggable genome, understudied kinase, dark kinome

## Abstract

**Simple Summary:**

Kinases are biomolecules that control cellular reactions essential for life. Disruptions in kinase expression and activity lead to the development of diseases such as cancer. A great deal of funding and scientific effort has been poured into kinase research, leading to great advancement. For example, a groundbreaking discovery was the development of imatinib, a kinase inhibitor, for the treatment of CML. Despite these discoveries, a large portion of the human kinome remains understudied. This review seeks to address this issue by discussing the understudied NEK family of kinases. We do this by discussing existing studies, analyzing correlations between the expression of NEK genes and patient survival outcomes in different cancers, discussing NEK mutations found in different cancer tissue types, and covering potential funding opportunities for those interested in pursuing NEK research.

**Abstract:**

The Never in Mitosis Gene A (NIMA)–related kinases (NEKs) are a group of serine/threonine kinases that are involved in a wide array of cellular processes including cell cycle regulation, DNA damage repair response (DDR), apoptosis, and microtubule organization. Recent studies have identified the involvement of NEK family members in various diseases such as autoimmune disorders, malignancies, and developmental defects. Despite the existing literature exemplifying the importance of the NEK family of kinases, this family of protein kinases remains understudied. This report seeks to provide a foundation for investigating the role of different NEKs in malignancies. We do this by evaluating the 11 NEK family kinase gene expression associations with patients’ overall survival (OS) from various cancers using the Kaplan–Meier Online Tool (KMPlotter) to correlate the relationship between mRNA expression of NEK1-11 in various cancers and patient survival. Furthermore, we use the Catalog of Somatic Mutations in Cancer (COSMIC) database to identify NEK family mutations in cancers of different tissues. Overall, the data suggest that the NEK family has varying associations with patient survival in different cancers with tumor-suppressive and tumor-promoting effects being tissue-dependent.

## 1. Introduction

The never in mitosis gene A (NIMA)-related kinases, commonly referred to as NEKs, are a family of 11 serine/threonine kinases (NEK1-NEK11) that was initially described in the fungus, *Aspergillus nidulans*, as an important initiator of mitosis [1,2]. The current literature has identified the involvement of NEK kinases in various essential cellular processes in mammals such as cell cycle regulation, DNA damage repair response (DDR), apoptosis, and primary ciliary function [3]. Additionally, disruptions and mutations in the NEK family of proteins have been linked to the development of various human cancers and diseases. For example, NEK10 is one of the kinases predicted to contain a driver mutation in a whole genome sequencing study of 210 primary tumors and immortalized cancer cell lines [4,5]. Another consequence of dysregulation of NEKs is illustrated in a study by Kenna et al., where there was a significant correlation in variant forms of NEK1 and the development of Amyotrophic Lateral Sclerosis (ALS), an autoimmune disease where there is selective degeneration of corticospinal tracts leading to paralysis and ultimately respiratory failure [6]. Lastly, a different study found that NEK4 knockdown in lymphoma promoted cellular microtubule stability in the presence of chemotherapeutic agents Taxol and Vincristine, making NEK4 a potentially selective marker for treating chemotherapy-resistant lymphoma based on patient-specific expression levels [7]. Despite some of the existing literature describing the role of NEKs in different aspects of cell biology and in the pathogenesis of various human diseases, the NEK family remains largely under-characterized. To exemplify the under-reported nature of the NEK family, a 2021 funding opportunity announcement by the NIH listed 9 out of 11 NEKs as understudied (NEKs 1, 3, 4, 5, 6, 7, 9, 10, 11). Oddly enough, the remaining members, NEK2 and NEK8, have had significantly less coverage in the recent literature compared to other kinases despite not making this list of understudied kinases. For example, PubMed searches for EGFR, a well-known tyrosine kinase, returned 107,410 results, while NEK2, being the most characterized NEK family member, returned 409 and NEK8 returned only 67. Because the NEK family of kinases regulate key cellular processes, how disruptions in their signaling is associated with disease development, and their status as under-characterized kinases, further research is warranted to investigate the detailed signaling mechanisms of NEKs and how they contribute to the physiologic and pathologic processes in biology. In this review, we briefly summarize key published details on each individual NEK and use the KMPlotter tool [8], a bioinformatics database designed to assess the correlations between gene expression (mRNA, miRNA, and protein) and survival in 30k+ patient samples from 21 tumor types, tissues to identify correlations in the overall survival (OS) of various cancers and mRNA expression of the 11 NEK kinases. Furthermore, we refer to the Catalogue of Somatic Mutations in Cancer (COSMIC) database and the NIH funding announcement RFA-RM-21-012, titled *Pilot Projects Investigating Understudied G Protein-Coupled Receptors, Ion Channels, and Protein Kinases*, to supplement KMPlot results with tissue mutation data and provide readers with opportunities for funding. Overall, our goal is to establish a foundation for NEK investigations in different malignancies and to highlight the potential of this under-characterized family of kinases.

## 2. Methods

### 2.1. Kaplan–Meier Plotter (KMPlot) Analysis

The NEK family mRNA expression levels and survival data correlations with different cancer types were generated by the KMPlot pan-cancer RNA-seq analysis. Groups of high vs. low NEK expression were algorithmically generated using the “Auto select best cutoff” option. Furthermore, all analyses were performed independently of disease subtype, grade, mutation burden, cellular content, or patient background. Parameters for discussions are *p*-value < 0.05 and Hazard Ratios > 1, for negative patient survival correlation, or <1, for positive patient survival correlation. Original KMPlot graphs are included with Appendix A and contain additional information such as the number of patients tracked (n) (Appendix A).

### 2.2. PubMed and Novelty Score Analysis

PubMed and Novelty scores were generated using the Pharos database, a user interface to the Knowledge Management Center (KMC) for the Illuminating the Druggable Genome (IDG) program funded by the National Institutes of Health (NIH) Common Fund [9]. The description for PubMed scores is: “Jensen Lab generated fractional counting score for the prevalence of this target in PubMed articles”. The description for Novelty scores is: “Tin-X metric for the relative scarcity of specific publications for this target”. To briefly summarize for readers to quickly interpret scores presented, higher PubMed scores represent a higher availability of publications for the kinase of interest and lower log Novelty scores (more negative values) represent a lower novelty for the kinase of interest. Additionally, the basis for describing NEK members as “2021 NIH designated understudied kinase” is based on the funding announcement RFA-RM-21-012, titled *Pilot Projects Investigating Understudied G Protein-Coupled Receptors, Ion Channels, and Protein Kinases*.

### 2.3. Catalogue of Somatic Mutations in Cancer (COSMIC) Analysis

Percent mutated values per cancerous tissue type were collected for each NEK family kinase from the COSMIC database and organized by each NEK family kinase (cancer.sanger.ac.uk) [10].

### 2.4. Gene Expression Profiling Interactive Analysis (GEPIA)

Comparisons of mRNA expression between normal and cancer tissues were generated using the GEPIA database for Appendix A. The *y*-axis unit is log_2_(TPM + 1) where TPM represents transcripts per million. The following is a list of tumor abbreviations for the *x*-axis labels: Adrenocortical carcinoma (ACC), bladder urothelial carcinoma (BLCA), breast invasive carcinoma (BRCA), cervical squamous cell carcinoma and endocervical adenocarcinoma (CESC), cholangio carcinoma (CHOL), colon adenocarcinoma (COAD), lymphoid neoplasm diffuse large b-cell lymphoma (DLBC), esophageal carcinoma (ESCA), glioblastoma multiforme (GBM), head and neck squamous cell carcinoma (HNSC), kidney chromophobe (KICH), kidney renal clear cell carcinoma (KIRC), kidney renal papillary cell carcinoma (KIRP), acute myeloid leukemia (LAML), brain lower grade glioma (LGG), liver hepatocellular carcinoma (LIHC), lung adenocarcinoma (LUAD), lung squamous cell carcinoma (LUSC), mesothelioma (MESO), ovarian serous cystadenocarcinoma (OV), pancreatic adenocarcinoma (PAAD), pheochromocytoma and paraganglioma (PCPG), prostate adenocarcinoma (PRAD), rectum adenocarcinoma (READ), sarcoma (SARC), skin cutaneous melanoma (SKCM), stomach adenocarcinoma (STAD), testicular germ cell tumors (TGCT), thyroid carcinoma (THCA), thymoma (THYM), uterine corpus endometrial carcinoma (UCEC), uterine carcinosarcoma (UCS), and uveal melanoma (UVM).

### 2.5. Kinase 3D Structure

X-ray crystal structures have been solved for three of the eleven NEK family members: NEK1 [2], NEK2 [11], and NEK7 [12] (Appendix A). In addition to their release and description in the primary literature, the 3D coordinates for these proteins can be found in the Protein Data Bank (PDB) [8]. As expected, they all display the typical overall bi-lobal architecture of protein kinases. As further interrogation of the NEK family takes place, a detailed molecular understanding of the interactions between NEK inhibitors and the NEK being studied will be important to drive improvements in potency and selectivity. The availability of structures for NEK1, NEK2, and NEK7 will facilitate the development of high-quality homology models of the other NEK family members. 

## 3. NEK1

NEK1 is highly expressed in meiotic cells and is involved in DNA repair processes, cell cycle progression, and ciliary function. Located on chromosome 4, NEK1 can be found in the cytoplasm, cilia, centrosome, and, during DNA damage response processes, the nucleus. Dysfunctions of NEK1 have been associated with the development and progression of Amyotrophic Lateral Sclerosis (ALS), Polycystic Kidney Disease (PKD), and cancers such as neural gliomas, Wilms’ tumor, prostate cancer, and thyroid cancer [13,14,15,16]. Furthermore, NEK1 is a 2021 NIH designated understudied kinase. KMPlotter analysis of NEK1 expression reveals positive patient survival correlations with esophageal squamous cell carcinoma, kidney renal cell carcinoma, kidney renal papillary carcinoma, pancreatic ductal carcinoma, and rectum adenocarcinoma (HR < 1, *p*-value < 0.05) and a negative correlation with survival in thyroid carcinoma (HR = 3.26, *p* < 0.05) (Table 1). COSMIC analysis reveals NEK1 mutations to be highly associated (greater than 7% of tissues with NEK1 mutations) with endometrial, liver, ovarian, pancreatic, and skin cancers, with 24.82% of mutations being missense substitutions.

## 4. NEK2

NEK2 is the most studied kinase in the NEK family, with more than 450 PubMed search results. Located on chromosome 1 along with NEK7, NEK2 has been found to have three splice variants, NEK2A, NEK2B, and NEK2C, that are involved in the regulation of cell cycling, centrosome integrity, cilia activity, and DNA splicing. Improper expression and activity of NEK2 has been associated with the development of many cancers, such as lung cancer, hepatocellular carcinoma, multiple myeloma, and pancreatic cancer [17,18,19,20,21]. KMPlotter analysis of NEK2 expression reveals positive patient survival correlations with esophageal squamous cell carcinoma, ovarian cancer, and thymomas (HR < 1, *p*-value < 0.05) and negative correlations with survival in esophageal adenocarcinoma, head and neck squamous cell carcinoma, kidney renal cell carcinoma, kidney renal papillary carcinoma, liver hepatocellular carcinoma, lung adenocarcinoma, pancreatic ductal carcinoma, sarcoma, and thyroid carcinoma (HR > 1, *p*-value < 0.05) (Table 1). Of the tissues tabulated in the COSMIC database, endometrial tissues demonstrated the highest percentage of NEK2 mutations (3.31%, *n* = 786).

## 5. NEK3

NEK3 is one of the least investigated members of the NEK family, with PubMed searches returning only 31 results. Unsurprisingly, NEK3 is a 2021 NIH designated understudied kinase. Located on chromosome 13 near NEK5, NEK3 has been implicated as a component of DNA damage repair processes and cell cycle regulation [22,23]. Dysregulations of NEK3 are associated with gastric, breast, and prostate cancers [12,24,25]. KMPlotter analysis of NEK3 expression reveals positive patient survival correlations with esophageal squamous cell carcinoma, pancreatic ductal carcinoma, sarcoma, stomach adenocarcinoma, and thyroid carcinoma (HR < 1, *p*-value < 0.05) and negative survival correlation with esophageal adenocarcinoma, kidney renal cell carcinoma, lung squamous cell carcinoma, ovarian cancer, and pheochromocytoma/paraganglioma (HR > 1, *p*-value < 0.05) (Table 1). Per COSMIC analysis, pancreatic tissues had the highest NEK3 mutation rate of 3.23% (*n* = 2011). 

## 6. NEK4

NEK4 is one of the lesser studied members of the NEK family, with PubMed searches returning only 36 results. Unsurprisingly, NEK4 is a 2021 NIH designated understudied kinase. Located on chromosome 3 with NEK10 and NEK11, NEK4 has two known major isoforms, NEK4.1 and NEK4.2. The NEK4 isoforms are believed to play a role in microtubule stabilization, cilia function, and DNA damage response [7,26,27,28]. Large scale genome analysis identifies NEK4 as a candidate gene for neurological conditions such as ADHD, schizophrenia, and bipolar disorder [29,30,31]. Furthermore, NEK4 has been found to be upregulated in lung and colon cancer tumors [32]. KMPlotter analysis of NEK4 expression reveals positive patient survival correlations with kidney renal cell carcinoma, rectum adenocarcinoma, and uterine corpus endometrial carcinoma (HR < 1, *p*-value < 0.05) and negative correlations with survival in liver hepatocellular carcinoma and sarcoma (HR > 1, *p*-value < 0.05) (Table 1). Additionally, COSMIC analysis reveals relatively low rates of NEK4 mutation across the majority of tissues, with the highest mutation rates being seen in placental (6.45%, n = 31) and skin (3.03%, n = 1786) cancerous tissues.

## 7. NEK5

NEK5 is one of the least studied members of the NEK family, with PubMed searches returning only 24 results. Unsurprisingly, NEK5 is a 2021 NIH designated understudied kinase. Located near NEK3 on chromosome 13, NEK5 has been reported to play a role in centrosome regulation, DNA damage response, and mitochondria function [33,34]. Recently, NEK5 has been implicated by our group and others as an important driver in breast cancer migration, proliferation, and the epithelial-to-mesenchymal transition (EMT) process [35,36,37]. KMPlotter analysis of NEK5 expression reveals positive patient survival correlations with bladder carcinoma, kidney renal cell carcinoma, liver hepatocellular carcinoma, pancreatic ductal carcinoma, stomach adenocarcinoma, thymoma, and uterine corpus endometrial carcinoma (HR < 1, *p*-value < 0.05) and negative correlations with survival in esophageal adenocarcinoma and thyroid carcinoma (HR > 1, *p*-value < 0.05) (Table 1). COSMIC analysis shows that NEK5 mutations were relatively low in most tissues, with the highest levels found in skin (6.91%, n = 1786) and endometrial (3.94%, n = 786) cancerous tissues.

## 8. NEK6

NEK6 is a kinase that can be found in the cytoplasm, nucleus, mitotic spindle, and centrosome. Located on chromosome 9, NEK6 has been demonstrated to regulate mitotic spindle and kinetochore fiber formation, the metaphase–anaphase transition, and cytokinesis, and it is involved in DNA damage repair. Additionally, activation of NEK6 has been demonstrated to occur from direct phosphorylation by NEK9 [38,39,40]. Aberrant NEK6 expression and function has been associated with the malignant progression of cancers such as hepatocellular carcinoma, gastric cancer, prostate cancer, and colorectal cancer [41,42,43,44]. NEK6 is also a 2021 NIH designated understudied kinase. KMPlotter analysis of NEK6 expression reveals positive patient survival correlations with esophageal adenocarcinoma, kidney renal cell carcinoma, rectum adenocarcinoma, and uterine corpus endometrial carcinoma (HR < 1, *p*-value < 0.05) and negative correlations with survival in bladder carcinoma, cervical squamous cell carcinoma, head and neck squamous cell carcinoma, liver hepatocellular carcinoma, lung squamous cell carcinoma, ovarian cancer, pancreatic ductal carcinoma, and sarcoma (HR > 1, *p*-value < 0.05) (Table 1). High levels of NEK6 expression are associated with treatment resistance and poor outcomes in patients with serous ovarian cancer by generating pro-survival cues and protection from degradation [45]. COSMIC analysis revealed liver tissues had the most NEK6 mutations (4.55%, n = 2286).

## 9. NEK7

NEK7 is a primarily nuclear kinase and is located on chromosome 1 alongside NEK2. Similar to NEK6, NEK7 activation occurs from direct phosphorylation by NEK9. Once activated, NEK7 functions to regulate the integrity of telomeres by providing them with a protective effect from oxidative DNA damage [46]. More recently, NEK7 has been identified as a mediator of NLRP3 inflammasome activation, an innate immune response pathway that promotes inflammation and apoptosis [47,48,49,50]. Because of NEK7’s role in mediating the inflammasome response, it is associated with the inflammatory responses of conditions such as bacterial infections, gouty arthritis, diabetes, arterial disease, and inflammatory bowel disease [48,51,52,53,54,55]. Additionally, NEK7 has been associated with the disease progression of malignancies such as gastric cancer, pancreatic cancer, non-small cell lung cancers, and gallbladder carcinoma [56,57,58,59]. Furthermore, NEK7 is a 2021 NIH designated understudied kinase. KMPlotter analysis of NEK7 expression reveals positive patient survival correlations with head and neck squamous cell carcinoma, kidney renal cell carcinoma, and sarcoma (HR < 1, *p*-value < 0.05) and negative correlations with survival in kidney renal papillary carcinoma, liver hepatocellular carcinoma, pancreatic ductal carcinoma, pheochromocytoma/paraganglioma, rectum adenocarcinoma, stomach adenocarcinoma, and thyroid carcinoma (HR > 1, *p*-value < 0.05) (Table 1). Per COSMIC analysis, NEK7 mutations were most common in cancers of ovarian (4.86%, *n* = 967) and liver (6.73%, *n* = 2286) tissues.

## 10. NEK8

NEK8 can primarily be found on spindle poles, the centrosome, and in the cytoplasm. Located on chromosome 17, NEK8 has been demonstrated to be a ciliary kinase that has an important role in DNA damage response. Because of its status as a ciliary kinase, abnormal NEK8 activity is associated with renal ciliopathies, seen in Polycystic Kidney Disease and nephronophthisis, and developmental left–right asymmetry defects [60,61]. In addition to ciliopathies, abnormal NEK8 expression and activity is associated with malignancies such as gliomas, gastric cancer, and breast cancer [62,63,64]. KMPlotter analysis of NEK8 expression reveals positive patient survival correlations with bladder carcinoma, cervical squamous cell carcinoma, head and neck squamous cell carcinoma, kidney renal papillary carcinoma, lung adenocarcinoma, pancreatic ductal carcinoma, and pheochromocytoma/paraganglioma (HR < 1, *p*-value < 0.05) and negative correlations with survival in kidney renal cell carcinoma (HR > 1, *p*-value < 0.05) (Table 1). COSMIC analysis shows the highest rates of tissue mutation in placental (3.23%, n = 31), skin (3.2%, n = 1786), and stomach (3.14%, n = 1367) cancerous tissues.

## 11. NEK9

NEK9 is located on chromosome 14 and has been shown to be a direct activator of NEK6 and NEK7 by phosphorylation. It is primarily found on spindle poles and centrosomes and its major functions involve mediating centrosome organization and other mitotic processes [65,66]. NEK9 mutations have been linked to pathologies such as nevus comedonicus and lethal skeletal dysplasia [67,68]. Additionally, malignancies associated with abnormal NEK9 expression and activity are gastric cancer, breast cancer, and glioblastoma multiforme [69,70,71]. Furthermore, NEK9 is a 2021 NIH designated understudied kinase. KMPlotter analysis of NEK9 expression reveals positive patient survival correlations with esophageal squamous cell carcinoma, kidney renal cell carcinoma, lung squamous cell carcinoma, pancreatic ductal carcinoma, and uterine corpus endometrial carcinoma (HR < 1, *p*-value < 0.05) and negative correlations with survival in bladder carcinoma and stomach adenocarcinoma (HR > 1, *p*-value < 0.05) (Table 1). Per COSMIC analysis, NEK9 mutations were most common in placental (6.45%, n = 31) and skin (5.67%, n = 1786) cancer tissues.

## 12. NEK10

NEK10 is located on chromosome 3, along with NEK4 and NEK11, and is associated with cellular processes involving mitochondrial metabolism and DNA damage response [72,73]. Abnormalities with NEK10 expression or function have been associated with ciliary dysfunctions and breast cancer [74,75,76,77,78]. NEK10 is among the least reported NEK family members, with only 26 PubMed search results, and is a 2021 NIH designated understudied kinase. KMPlotter analysis of NEK10 expression reveals positive patient survival correlations with breast cancer, kidney renal papillary carcinoma, pancreatic ductal carcinoma, thymoma, and uterine corpus endometrial carcinoma (HR < 1, *p*-value < 0.05) and negative correlations with survival in kidney renal cell carcinoma, stomach adenocarcinoma, and thyroid carcinoma (HR > 1, *p*-value < 0.05) (Table 1). COSMIC analysis reveals NEK10 mutations are most present in endometrial (7.12%, *n* = 786), liver (10.63%, *n* = 2286), ovarian (7.03%, *n* = 967), pancreatic (7.64%, *n* = 2011), and skin (7.1%, *n* = 1786) cancers, with 21.78% of all mutations being missense substitutions. 

## 13. NEK11

NEK 11 is located on chromosome 3, along with NEK4 and NEK10, and has been linked with DNA replication and DNA damage response processes [79,80]. Exome sequencing analysis identifies a potential link between NEK11 and ketotic hypoglycemia and melanoma, but additional validation studies must be performed to confirm this finding [81,82]. NEK11 is the least reported NEK family member, with only 18 PubMed search results, and, understandably, is part of the list of 2021 NIH designated understudied kinases. KMPlotter analysis of NEK11 expression reveals positive patient survival correlations with breast cancer, kidney renal papillary carcinoma, pancreatic ductal carcinoma, thymoma, and uterine corpus endometrial carcinoma (HR < 1, *p*-value < 0.05) and negative correlations with survival in kidney renal cell carcinoma, stomach adenocarcinoma, and thyroid carcinoma (HR > 1, *p*-value < 0.05) (Table 1). Downregulation of NEK11, specifically in ovarian cancer, is associated with drug resistance and poor patient survival. COSMIC analysis reveals NEK11 mutations are most present in breast (7.05%, *n* = 2908), liver (12.38%, *n* = 2286), esophageal (7.33%, *n* = 1596), pancreatic (7.76%, *n* = 2011), penial (10%, *n* = 10), and prostate (7.74%, *n* = 1990) cancers, with 15.50% of all mutations being missense substitutions.

## 14. Discussion

Although understudied, a body of work has emerged that demonstrates the physiological roles for NEKs in the regulation of the cell cycle, mitosis, centrosome disjunction, cilia formation, and DNA damage repair. Additionally, dysregulations of NEK family members have been demonstrated to be associated with pathologies such as neurodegenerative diseases, dystrophies, ciliopathies, and malignancies. In this manuscript, we review the existing literature to obtain a broad overview of the individual kinases in the NEK family. Furthermore, we use the KMPlotter database to correlate mRNA levels of NEK family members with patient survival outcomes in different cancers and the COSMIC database to briefly discuss tissues with common NEK mutations. Data analysis reveals a complicated relationship between individual members of the NEK family and cancers of different tissue types. To begin with, all NEK family kinases have both positive (HR < 1) and negative (HR > 1) survival correlations for patient survival outcomes, depending on the cancer type. Although the majority of cancers have both positive (HR < 1) and negative (HR > 1) patient survival correlations, depending on the context of which NEK family member is analyzed, only positive patient survival correlations are seen with breast cancer, esophageal squamous cell carcinoma, thyroid carcinoma, and uterine corpus endometrial carcinoma (excluding non-statistically significant relationships). Additionally, testicular germ cell tumors have no statistically significant correlations with any NEK family member. This demonstrates that members of the NEK family cannot simply be characterized as an overall tumor-suppressor or tumor-promoting gene because the relationship is dependent on the disease context. That being said, the KMPlot data suggest that the NEK family overall may be more inclined to have tumor-suppressive roles in breast, esophageal, thyroid, uterine, and endometrial tissues. Furthermore, NEK1 and NEK8 further support this inclination by being predominantly associated with improved survival outcomes (positive-to-negative ratios of 5:1 for NEK1 and 7:1 for NEK8) despite the other NEK members having a more balanced split between positive and negative survival correlations in different cancers.

NEK2 is one of the better studied kinases in the NEK family and has been reported to be associated with disease progression for lung cancer, hepatocellular carcinoma, and pancreatic cancer. Interestingly, the KMPlot data seem to form a consensus with the existing literature by showing hazard ratios that are associated with poor survival for these three cancer types. A different study, by Uddin et al., demonstrated that NEK2 expression is associated with stemness marker ALDH1A1 in treatment-resistant ovarian cancer [83]. Although this study suggests a disease-promoting role for NEK2 in ovarian cancer, the KMPlot data demonstrate a conflicting view by showing a beneficial hazard ratio for patients with higher NEK2 levels. These contradictory results highlight a weakness with relying on bioinformatics data alone and further emphasize the importance of performing additional validation experiments. Altogether, KMPlot and the literature for NEK2 has revealed it to be a potential target for small molecule therapeutics in recurring ovarian and other aggressive cancers.

NEK5 is an understudied kinase and is part of the NIH designated IDG kinase list. We and others recently reported NEK5 as a promoter of EMT, migration, and proliferation in Triple Negative Breast Cancers (TNBC) [35,36,37]. Interestingly, despite the literature and our experimental results demonstrating NEK5 to be a driver for TNBC function and progression, there is no statistical correlation between NEK5 mRNA expression and breast cancer survival based on the KMPlot data. We believe that this discrepancy is the result of one or a combination of the two following reasons. Firstly, mRNA expression does not always necessarily directly translate to protein levels and function. If this were the case, KMPlot’s mRNA data would not accurately reflect NEK5’s levels and activity in breast cancer cells. To overcome this limitation, additional validations, by techniques such as Western blotting and microscopy, must be performed. Secondly, this analysis was performed for all breast cancers without consideration for tumor subtype, grade, stage, and patient treatment status. Breast cancer is a complex disease with distinct molecular subtypes. The studies for NEK5 by our group and others were performed using basal subtype cell line models, while the KMPlot analysis included both the basal and the biologically different luminal subtypes. Additionally, our study utilized a novel patient-derived xenograft cell line TU-BcX-4IC. This model was derived by our group from a patient with metaplastic carcinoma post neoadjuvant therapy. This is an important point to consider because therapy may select for specific subpopulations within the tumor microenvironment, resulting in a fundamental change in tumor biology. Although our decision to analyze breast cancer overall simplifies the results of our NEK family review, we would like to also point out the possibility of type 2 errors (missing a correlation that does exist) due to the clumping of distinct breast cancer categories into one group. Altogether, NEK5 and breast cancer survival correlations is an example of the limitations associated with data generated by KMPlotter.

NEK6, NEK7, and NEK9 have been demonstrated to be members of the NEK family that interact with each other due to NEK9’s role as an activator, by phosphorylation, of NEK6 and NEK7. Despite this existing relationship, NEK6, NEK7, and NEK9, for the most part, do not share similar patient survival correlations with each other. Interestingly, the KMPlot data do show all three of these kinases to be positively correlated with patient survival outcomes in kidney renal cell carcinoma. Unfortunately, due to the understudied nature of this group of NEK kinases, there is no existing literature that covers this interaction and the resulting effects on kidney renal cell carcinoma biology. Nonetheless, the KMPlot data suggest potential avenues for exploring the tumor-suppressive qualities of NEK6, NEK7, and NEK9 in kidney renal cell carcinoma. Additionally, NEK6 and NEK9 both have negative correlations for patient survival outcomes in bladder carcinoma and positive correlations for patient survival outcomes in uterine corpus endometrial carcinoma. This is further supported by GEPIA data, as uterine corpus endometrial carcinoma has the lowest tumor:normal tissue expression ratios of NEK7 and NEK9 (Table 2). Furthermore, both NEK6 and NEK7 have negative correlations for patient survival outcomes in liver hepatocellular carcinoma and pancreatic ductal carcinoma. Overall, the relationships and interactions between NEK6, NEK7, and NEK9 may prove to be interesting avenues of research. Furthermore, investigations on the effects between NEK6, NEK7, and NEK9 in kidney renal cell carcinoma, and whether the effects are additive or synergistic, could also reveal interesting results.

Animal model knockouts of the NEK family have also given additional insight into the role of these kinases. NEK1 knockout mice result in the failure of cell cycle checkpoint arrest in response to DNA damage, resulting in genomic instability and lymphoma development [1,10]. NEK2, the most studied family member, has commercially available mouse knockout models available. Studies using these animal models have identified a critical role for B cell tumor development and progression [2]. Knockout of NEK6 in mice led to the promotion of cardiac hypertrophy, dysfunction, and fibrosis through activation of AKT signaling, though no tumor studies were performed [3]. NEK7 knockout mouse studies showed a time-dependent effect on development. NEK7 absence during embryogenesis led to lethality, while post-natal absence resulted in severe growth retardation. Analysis of embryonic fibroblasts demonstrated an increased tendency for micronuclei formation and failure of cytokinesis [4]. Although NEK9 knockout mice do not exist, studies of NEK9 mutant mice resulted in impaired kidney cilia formation that led to renal deficiencies [5]. There are no current knockout mouse model studies for the remaining NEK family kinases.

In addition to reviewing the existing literature and analyzing the mRNA expression and patient survival outcome correlations in the KMPlot database, we used the Catalogue of Somatic Mutations in Cancer (COSMIC) database to explore NEK mutations in different tissue types. Our initial COSMIC analysis revealed that the NEK family kinase mutations are variable across various cancer types, with nearly all tissue types tested having mutations of all the NEK family kinases. To narrow down the list of interesting mutation correlations, we limit our discussions to the top mutated tissue for each kinase member of the NEK family. Furthermore, we also bring up tissues that have a >7% NEK mutation rate. Liver, ovarian, and pancreatic cancer tissues had multiple NEKs with greater than 7% mutated. Additionally, following a stage 4 diagnosis, the 5-year survival rate of these cancers decreases precipitously to below 20%. In addition to NEK family kinase members emerging as therapeutic targets in cancers due to their roles in treatment resistance, metastasis, and poor patient prognosis, the COSMIC data suggest that additional studies in NEK mutations may yield interesting and complimentary results.

The human genome encodes for more than 500 kinases, enzymes that regulate biological processes by phosphorylation [84]. Kinase inhibition has yielded promising results for treating diseases. For example, imatinib, a tyrosine kinase inhibitor that blocks Bcr-Abl, has greatly increased the 10-year survival rate of chronic myelogenous leukemia (CML) from <15% to >80% [85,86]. Despite the amount of effort and funding poured into kinase research, 30% of human kinases remain understudied [87]. To address this gap, the NIH launched the Illuminating the Druggable Genome (IDG) program which includes the 2021 NIH designated understudied kinase funding announcement RFA-RM-21-012, titled *Pilot Projects Investigating Understudied G Protein-Coupled Receptors, Ion Channels, and Protein Kinases*. NEKs are an understudied family of kinases. Of the 11 members, 9 (NEKs 1, 3, 4, 5, 6, 7, 9, 10, 11) are eligible for NIH RFA-RM-21-012 funding (Table 2). Interestingly, NEK8 coverage is comparable to other members of this family but it failed to make the list. Additionally, NEK2 is the best studied member of the NEK family but has significantly less coverage in the recent literature compared to many other kinases. For example, NEK2 has a PubMed score of 182.71, while AKT1 has a PubMed score of 29,775, a 100-fold difference in magnitude (Table 2).

The NEK family is a family of 11 kinases that are relatively poorly characterized. What is known is that members of this kinase family are involved in a variety of essential cellular processes such as cell cycle regulation, DNA damage repair response (DDR), apoptosis, and microtubule organization. Furthermore, emerging studies implicate NEK members in the development or progression of different cancers. The broadly ranging results suggest that the relationships between NEKs and cancer outcomes are variable and depend on disease context. Limitations in this report include looking at overall patient survival without separating by disease stage, grade, treatment status, or subtype. Nevertheless, the work reported here establishes a foundation for future investigations of NEKs and cancer by summarizing correlations between mRNA expression levels of different NEKs and patient survival outcomes in different cancers. Additionally, we highlight an NIH funding opportunity for potential investigators interested in pursuing a project with the NEK family. We believe that this review reiterates the importance of the NEK family, provides a guide that can help scientists initiate NEK biology and chemistry discovery projects in areas of interest, and makes it clear that there are many unanswered questions on the functions of all NEKs. Based on the volume of unanswered questions that we believe is due to the lack of attention paid to this family, it will be important to perform additional validation studies that use NEK inhibitors, and multiple cell lines (with and without NEK family member knockdown/knockout) that can provide additional context to the positive or negative data seen in the KMPlot results. Further study will likely validate at least a subset of the NEK family for fully fledged drug discovery efforts, and may lead to the identification of new medicines for oncologic indications and also other diseases.

## Figures and Tables

**Table 1 cancers-15-02067-t001:** Hazard ratios (HR) and associated confidence intervals (CI) comparing correlations between mRNA expression of NEK family kinases and patient survival outcomes for different cancers. Statistically significant (*p* < 0.05) correlations are colored green for positive survival benefit and red for negative survival benefit, non-statistically significant correlations are colored black, and correlations with insufficient power (low n) are shown in blue.

Tumor Type	NEK1 HR	NEK2 HR	NEK3 HR	NEK4 HR	NEK5 HR	NEK6 HR	NEK7 HR	NEK8 HR	NEK9 HR	NEK10 HR	NEK11 HR
Bladder Carcinoma	1.17 (0.86–1.59)	0.83 (0.59–1.18)	0.75 (0.56–1.01)	0.78 (0.58–1.04)	0.69 (0.52–0.93)	1.73 (1.16–2.57)	1.23 ( 0.91–1.66)	0.66 (0.49–0.89)	1.45 (1.08–1.94)	0.86 (0.62–1.18)	0.86 (0.62–1.18)
Breast Cancer	0.72 ( 0.52–1)	1.28 ( 0.91–1.8)	1.19 (0.82–1.74)	0.82 (0.59–1.13)	0.74 (0.54–1.02)	1.35 (0.98–1.87)	0.82 (0.59–1.13)	0.83 (0.6–1.14)	0.75 (0.53–1.06)	0.46 (0.29–0.72)	0.46 (0.29–0.72)
Cervical SCC	1.37 (0.76–2.47)	1.33 (0.83–2.13)	1.49 (0.92–2.43)	1.42 (0.8–2.51)	0.75 (0.47–1.19)	1.87 (1.15–3.05)	1.33 (0.82–2.16)	0.58 (0.34–1)	1.58 (0.99–2.52)	1.35 (0.84–2.17)	1.35 (0.84–2.17)
Esophageal Adenocarcinoma	0.5 (0.24–1.02)	2.76 (1.07–7.15)	2.87 (1.12–7.38)	0.56 (0.26–1.23)	2.28 (1.15–4.55)	0.4 (0.21–0.76)	1.8 (0.88–3.71)	0.52 (0.26–1.05)	0.58 (0.3–1.11)	1.81 (0.76–4.35)	1.81 (0.76–4.35)
Esophageal SCC	0.06 (0.01–0.43)	0.4 (0.18–0.92)	0.44 (0.2–0.99)	0.55 (0.22–1.38)	1.64 (0.74–3.65)	0.5 (0.21–1.17)	1.38 (0.55–3.48)	0.57 (0.23–1.39)	0.3 (0.13–0.7)	0.54 (0.23–1.27)	0.54 (0.23–1.27)
Head and neck SCC	0.86 (0.65–1.15)	1.44 (1.08–1.92)	0.82 (0.61–1.11)	0.76 (0.57–1.02)	0.9 (0.68–1.18)	1.46 (1.05–2.03)	0.74 (0.57–0.97)	0.67 (0.51–0.89)	1.16 (0.89–1.53)	1.19 (0.89–1.59)	1.19 (0.89–1.59)
Kidney RCC	0.52 (0.4–0.72)	2.46 (1.81–3.34)	1.7 (1.25–2.3)	0.6 (0.45–0.81)	0.73 (0.53–1)	0.41 (0.28–0.62)	0.58 (0.42–0.79)	1.65 (1.21–2.25)	0.67 (0.5–0.91)	1.69 (1.25–2.29)	1.69 (1.25–2.29)
Kidney Renal Papillary Carcinoma	0.46 (0.24–0.86)	4.71 (2.59–8.56)	0.64 (0.36–1.17)	1.6 (0.76–3.35)	1.5 (0.81–2.75)	0.64 (0.36–1.17)	2.07 (1.14–3.78)	0.49 (0.27–0.91)	0.76 (0.39–1.45)	0.45 (0.25–0.81)	0.45 (0.25–0.81)
Liver Hepatocellular Carcinoma	0.89 (0.63–1.26)	2.14 (1.52–3.03)	0.82 (0.57–1.18)	1.66 (1.13–2.44)	0.61 (0.42–0.86)	1.75 (1.2–2.54)	1.55 (1.06–2.25)	1.38 (0.97–1.96)	1.29 (0.88–1.9)	1.33 (0.91–1.95)	1.33 (0.91–1.95)
Lung Adenocarcinoma	1.24 (0.88–1.74)	1.84 (1.36–2.49)	0.77 (0.57–1.05)	1.22 (0.88–1.69)	0.77 ( 0.57–1.05)	1.3 ( 0.96–1.77)	0.79 (0.58–1.06)	0.64 (0.48–0.87)	0.79 (0.58–1.09)	0.83 (0.62–1.11)	0.83 (0.62–1.11)
Lung SCC	1.18 ( 0.89–1.56)	0.8 (0.6–1.07)	1.36 (1.03–1.79)	0.8 (0.6–1.05)	1.29 (0.94–1.77)	1.55 (1.18–2.04)	0.79 (0.6–1.04)	1.24 (0.93–1.67)	0.66 (0.48–0.9)	1.19 (0.88–1.61)	1.19 (0.88–1.61)
Ovarian Cancer	0.85 (0.62–1.14)	0.74 (0.55–0.98)	1.46 (1.12–1.91)	0.76 (0.57–1.01)	1.22 (0.94–1.58)	1.34 (1–1.79)	1.15 (0.87–1.51)	1.19 (0.92–1.54)	1.25 (0.94–1.66)	1.33 (1–1.77)	1.33 (1–1.77)
Pancreatic Ductal Carcinoma	0.55 (0.36–0.83)	1.99 (1.31–3.03)	0.51 (0.34–0.78)	1.26 (0.84–1.9)	0.59 (0.38–0.9)	2.57 (1.43–4.63)	1.97 (1.29–3.01)	0.43 ( 0.25–0.72)	0.42 (0.25–0.72)	0.54 (0.36–0.82)	0.54 (0.36–0.82)
Pheochromocytoma & Paraganglioma	2.7 (0.49–14.95)	3.91 (0.7–22)	5.33 (0.97–29.43)	4.46 (0.75–26.7)	0.24 ( 0.04–1.36)	0.39 (0.07–2.18)	4.76 (0.87–26.22)	0.07 (0.01–0.57)	4.5 (0.75–26.98)	587,911,007.09 (0–inf)	587,911,007.09 (0–inf)
Rectum Adenocarcinoma	0.39 (0.18–0.86)	0.75 (0.34–1.7)	0.5 (0.22–1.12)	0.26 (0.12–0.57)	1.75 (0.81–3.79)	0.31 (0.09–1.05)	2.17 (1–4.71)	2.1 (0.84–5.26)	2.18 (0.65–7.29)	0.45 (0.17–1.19)	0.45 (0.17–1.19)
Sarcoma	0.7 (0.46–1.05)	1.85 (1.2–2.86)	0.55 (0.35–0.86)	1.88 (1.15–3.08)	0.57 (0.37–0.89)	2.03 (1.34–3.09)	0.64 (0.41–1)	0.72 (0.48–1.07)	0.77 (0.48–1.22)	0.72 (0.48–1.07)	0.72 (0.48–1.07)
Stomach Adenocarcinoma	1.23 (0.88–1.7)	0.73 (0.52–1.01)	0.65 (0.43–0.98)	0.77 (0.54–1.11)	0.56 (0.38–0.82)	1.24 (0.84–1.82)	1.61 (1.14–2.27)	0.76 (0.53–1.09)	1.42 (1.02–1.97)	1.69 (1.14–2.51)	1.69 (1.14–2.51)
Testicular Germ Cell tumor	0.18 (0.02–1.18)	0.14 (0.01–1.58)	0.17 (0.02–1.9)	2.93 (0.3–28.18)	0.18 (0.02–1.83)	2,023,412,437.18 (0-inf)	22,623,872.43 (0-inf)	2.44 (0.34–17.39)	0.17 (0.02–1.17)	0.14 (0.01–1.39)	0.14 (0.01–1.39)
Thymoma	1.98 (0.52–7.55)	0.1 (0.02–0.49)	3.06 (0.75–12.43)	0.33 (0.09–1.26)	0.2 (0.05–0.79)	2.55 ( 0.67–9.73)	2.56 (0.68–9.63)	815,958,421.87 (0- inf )	1.73 (0.46–6.47)	0.15 (0.03–0.73)	0.15 (0.03–0.73)
Thyroid Carcinoma	3.26 (1.05–10.13)	3.55 (1.14–11.03)	0.35 (0.12–1.01)	2.32 (0.87–6.24)	3.21 (1.2–8.56)	5.84 (0.77–44.24)	3.33 (1.15–9.58)	0.67 (0.24–1.85)	2.08 (0.59–7.32)	3.51 (1.31–9.4)	3.51 (1.31–9.4)
Uterine corpus endometrial carcinoma	0.74 (0.47–1.17)	1.43 (0.95–2.17)	1.32 (0.87–2)	0.54 (0.33–0.87)	0.5 (0.3–0.83)	0.58 (0.37–0.91)	1.49 (0.99–2.26)	1.42 (0.93–2.18)	0.6 ( 0.4–0.92)	0.63 (0.41–0.96)	0.63 (0.41–0.96)

**Table 2 cancers-15-02067-t002:** Scores from Pharos, the user interface to the Knowledge Management Center (KMC) for the Illuminating the Druggable Genome (IDG) program funded by the National Institutes of Health (NIH) Common Fund. EGFR, AKT1, and MAPK1 are included as better studied molecules for comparison with the NEK family of kinases. The PubMed score is described as “Jensen Lab generated fractional counting score for the prevalence of this target in PubMed articles”. The log Novelty Score is described as “TIN-X metric for the relative scarcity of specific publications for this target”. The Antibody Count is described as “Number of antibodies for this target listed in antibodypedia.com”. NEKs with an asterisk (*) denote 2021 (RFA-RM-21-012) NIH designated understudied kinases. Furthermore, this table contains a summary of information on each kinase member, including expression level comparisons between tumor and normal tissues from the GEPIA database (Appendix A).

Kinase	PubMed Score	Novelty Score (Log)	Antibody Count	Chromosome	Localization	Biological Processes	Associated Pathologies	Highest Tumor: Normal Tissue Expression Ratio by GEPIA	Lowest Tumor: Normal Tissue Expression Ratio by GEPIA
EGFR	15,684.43	−9.53	8544	7	Cytoplasm, plasma membrane	Cell division and survival	Fibrosis, atherosclerosis, adenocarcinoma of the lung, glioblastoma, head and neck tumors	Thymoma, 124.33-fold	Skin Cutaneous Melanoma, 0.10-fold
AKT1	29,775.56	−10.35	4668	14	Cytoplasm, nucleus	Cell growth, proliferation, and apoptosis	Proteus syndrome, diabetes, cancer	Thymoma, 1.36-fold	Adrenocortical carcinoma, 0.90-fold
MAPK1	1421.81	−7.4	1683	16	Cytoplasm, nucleus	Proliferation, differentiation, and transcription regulation	Alzheimer’s disease, Parkinson’s disease, ALS, cancer	Cholangio carcinoma, 1.72-fold	Adrenocortical carcinoma, 0.73-fold
NEK1 *	63.71	−4.1	125	4	Cytoplasm, nucleus	DNA damage repair	ALS, PKD, Wilm’s tumor, prostate cancer, thyroid cancer	Thymoma, 7.13-fold	Uterine Corpus Endometrial Carcinoma, 0.60-fold
NEK2	182.71	−5.12	423	1	Cytoplasm, nucleus	Cell cycling, cilia activity, DNA splicing	Lung cancer, hepatocellular carcinoma, multiple myeloma, pancreatic cancer	Cervical squamous cell carcinoma and endocervial adenocarcinoma, 38.58-fold	Acute Myeloid Leukemia, 0.46-fold
NEK3 *	10.99	−2.35	223	13	Cytoplasm, nucleus	DNA damage repair, cell cycling	Gastric cancer, breast cancer, prostate cancer	Thymoma, 2.74-fold	Kidney Chromophobe, 0.54-fold
NEK4 *	17.05	−2.85	192	3	Cytoplasm, nucleus	Cell Cycling, mitosis, DNA damage response	ADHD, schizophrenia, bipolar disorder, lung cancer, colon cancer	Thymoma, 2.88-fold	Testicular Germ Cell Tumors, 0.66-fold
NEK5 *	8.91	−1.7	145	13	Cytoplasm, nucleus	DNA damage response, mitochondria function	Breast cancer	Cholangio carcinoma, 4.64-fold	Lung squamous cell carcinoma, 0.30-fold
NEK6 *	44.06	−3.73	269	9	Cytoplasm, nucleus	DNA damage repair, cell cycling	Hepatocellular carcinoma, gastric cancer, prostate cancer, colorectal cancer	Skin Cutaneous Melanoma, 2.16-fold	Kidney Chromophobe, 0.41-fold
NEK7 *	41.34	−3.62	332	1	Cytoplasm, nucleus	Telomere integrity, NLRP3 inflammasome activation, inflammation, apoptosis	Gouty arthritis, diabetes, arterial disease, inflammatory bowel disease, gastric cancer, pancreatic cancer, non-small cell lung cancer, gallbladder carcinoma	Pancreatic adenocarcinoma, 1.74-fold	Uterine Corpus Endometrial Carcinoma, 0.69-fold
NEK8	137.44	−4.95	271	17	Cytoplasm, nucleus	Ciliary biogenesis and DNA damage response	Renal ciliopathies, polycystic kidney disease, nephronophthisis, left-right assymetry, gliomas, gastric cancer, breast cancer	Lymphoid Neoplasm Diffuse Large B-Cell Lymphoma, 3.12-fold	Kidney Chromophobe, 0.61-fold
NEK9 *	46.5	−3.74	446	14	Cytoplasm, nucleus	Centrosome organization and mitosis	Nevus comedonicus, lethal skeletal dysplasia, gastric cancer, breast cancer, glioblastoma	Thymoma, 1.59-fold	Uterine Corpus Endometrial Carcinoma, 0.69-fold
NEK10 *	9.38	−2.11	60	3	Cytoplasm, nucleus	Mitochondrial metabolism and DNA damage response	Ciliary dysfunction and breast cancer	Pancreatic adenocarcinoma, 3.21-fold. Acute Myeloid Leukemia may also be of interest due to 0 expression in normal tissues vs 0.53 in tumor	Testicular Germ Cell Tumors, 0.13-fold
NEK11 *	7.48	−2.08	244	3	Nucleus	DNA replication and DNA damage response	Ketotic hypoglycemia and melanoma	Cholangio carcinoma, 6.72-fold	Kidney Chromophobe, 0.15-fold

## Data Availability

The Kaplan–Meier Plotter database can be accessed at https://kmplot.com/analysis/. The data generated in this manuscript was based on access dates 21 November 2021. The parameters for this review are listed in the Methods section. The Catalogue of Somatic Mutations in Cancer (COSMIC) database can be accessed at https://cancer.sanger.ac.uh/cosmic. The values for Table 2 can be accessed at https://pharos.ncats.nih.gov with our access date being 21 November 2021. More information on the NIH funding opportunity RFA-RM-21-012 can be accessed at https://grants.nih.gov/grants/guide/rfa-files/RFA-RM-21-012.html with our access date being 21 November 2021.

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
