# Peer review of "NEK Family Review and Correlations with Patient Survival Outcomes in Various Cancer Types"

_cancers, 2023, doi:10.3390/cancers15072067_

Round 1

Reviewer 1 Report

Nguyen and group have studied the gene expression of eleven NIMA-related kinases in different cancers. Authors have studied overall survival using the Kaplan-Meier Online Tool (KM Plotter) to correlate the relationship between mRNA expression of NEK1-11 in various cancers and patient survival. Moreover, the mutation status of these genes was also studied using the COSMIC database. The study showed that the NEK family of proteins is associated with overall survival in cancer. This is mainly a bioinformatic study performed using various online databases of existing data. I feel this provides the importance of the family of NEK proteins. I suggest you include overall survival KM curves with cutoff and p-value so it will be easy to understand the OS. Moreover, you can always add mRNA expression graphs and compare normal vs cancer expression. It will be also interesting to add a correlation of NEK family genes with genes involved in cancer progression, metastasis, stemness, etc. This will make studying more important. Lastly, I feel it is important to compare this OS and mutation data with current literature and summarize recent reports. 

Reviewer 2 Report

In this manuscript, the authors intended to provide an overview of the literature related to the members of the NEK family to highlight the scarcity of studies dedicated to them.

Because all the proteins described in this review belong to the same family, the authors should first describe their common or specific traits, structure, signaling… A whole section should be dedicated to this issue. A scheme is needed.

Because each member of the NEK family is described separately, sections 3-13 are very repetitive in their organization. The authors should modify all these sections.

The authors should complement this review by addressing the existence of knockout animals that help to decipher physiological roles and the development of NEK inhibitors.

Minor concerns :

Table 1 : The table legend lacks explanations for the correlations highlighted in blue and black.

Reviewer 3 Report

There have not been many comprehensive reviews of NEK kinases published in recent years, and hardly any that cover all of them in one manuscript. This niche article provides a thorough overview of the currently known physiological role of the NEK kinase family and evaluates their potential role in various tumors, and also investigates possible links between NEK kinases and survival of cancer patient.

The article is well-written, based on publications in the field (which are properly cited).

I have just a few questions, and a recommendation:

What is the structure of NEK kinase proteins?

Intracellularly, where are the different NEK kinases localised? Are they soluble, membrane bound? It is also worth analysing their physiological tissue-specific expression in more detail.

The article lacks a summary table, which includes the main characteristics of NEK kinases (which are listed in the article) to help an easier overview. 

Round 2

Reviewer 2 Report

I would like to thank the authors for their clear answers as well as for the modifications made to the manuscript. 

Author Response

.